# Drug-Free Noninvasive Thermal Nerve Block: Validation of Sham Devices

**DOI:** 10.3390/brainsci13121718

**Published:** 2023-12-15

**Authors:** Michael A. Fishman, Ahish Chitneni, Alaa Abd-Elsayed, Samuel Grodofsky, Ashley M. Scherer, Brendan Schetzner, Malvina Klusek, Stephen R. Popielarski, Stephen Meloni, Steven Falowski, Philip Kim, Konstantin V. Slavin, Stephen D. Silberstein

**Affiliations:** 1Center for Pain Control PC, Lancaster, PA 19610, USA; mafishman@gmail.com (M.A.F.); ashleys@centerisp.com (A.M.S.); 2Department of Rehabilitation and Regenerative Medicine, New York-Presbyterian Hospital—Columbia and Cornell, New York, NY 10065, USA; 3Department of Anesthesia, Division of Pain Medicine, University of Wisconsin School of Medicine and Public Health, Madison, WI 53726, USA; alaaawny@hotmail.com; 4Philadelphia Smart Pain & Wellness, Bala Cynwyd, PA 19610, USA; sgrodo@phillysmartpain.com; 5Department of Anesthesiology, St Elizabeths Medical Center, Brighton, MA 02135, USA; brendan.schnetzer@gmail.com; 6Peconic Bay Medical Center/Northwell Health, New York, NY 10065, USA; mklusek@northwell.edu; 7Thermaquil, Inc., Philadelphia, PA 19610, USA; steve@thermaquil.com (S.R.P.); stephen.meloni@thermaquil.com (S.M.); 8Neurosurgical Associates of Lancaster, Lancaster, PA 19610, USA; sfalowski@gmail.com; 9Center for Interventional Pain Spine, LLC, Wilmington, DE 19803, USA; phshkim@yahoo.com; 10Department of Neurosurgery, University of Illinois at Chicago, Chicago, IL 60607, USA; kslavin@uic.edu; 11Neurology Service, Jesse Brown Veterans Administration Medical Center, Chicago, IL 60612, USA; 12Jefferson Headache Center, Thomas Jefferson University, Philadelphia, PA 19107, USA; stephen.silberstein@jefferson.edu

**Keywords:** sham device, chronic pain, thermal nerve block, drug-free nerve block

## Abstract

Headache is a leading cause of disability and suffering. One major challenge in developing device treatments is demonstrating their efficacy given devices’ often-high placebo rate. This paper reviews the importance of validating sham devices as part of finalizing the design for larger-scale prospective randomized controlled trials in patients with chronic headache as well as the results of a prospective, single-blind trial to validate two potential sham noninvasive thermal nerve block devices. Study participants were trained to self-administer thermal nerve block treatment using sham devices in an office visit. Two different sham systems with different temperature profiles were assessed. Devices were offered for patients to use daily at-home for one week to assess the durability of sham placebo effects before participants were given active treatment in a second office visit followed by another optional week of self-administered active treatment at-home use. Sham treatments reduced pain scores by an average of 31% from 6.0 ± 2.3 to 4.3 ± 3.3, including two participants who fell asleep during the in-office treatment and woke up with no pain, but whose pain recurred after returning home during at-home use of the sham system. In-office active treatments reduced pain scores by 52% from 6.7 ± 2.1 to 3.3 ± 2.9 with sustained pain relief during optional at-home use. Successful blinding for the study was confirmed with an ideal Bang’s Blinding Index of 0 and an ideal James’ Blinding Index of 1. Both the sham and active treatments were viewed by participants as highly credible, and credibility increased from the beginning to end of sham treatments on average.

## 1. Introduction

Chronic headaches, particularly migraine, are debilitating neurological conditions often refractory to medical management, leaving patients and physicians with few options for pain relief. A recent review of over 230 headache epidemiological studies found the average lifetime prevalence for any active primary headache disorder was 66.6%, and the prevalence for experiencing headache ≥ 15 days per month (chronic migraine (CM) and chronic-tension-type headache (CTTH)) was 4.5% [1]. Migraine alone causes more years lived with disability for 15–49-year-old men and women than any other condition and is the leading cause of disability for women overall [2]. The disease burden is especially high for patients who failed preventative treatments, with 74% of patients reporting spending time in darkness/isolation an average of 19 h/month, 83% reporting sleeping difficulty, and 85% reporting feelings of helplessness, depression, or not being understood [3]. Over 4 million headache patients utilize US emergency departments annually [4], and 73% of patients suffer from headache recurrence within 48 h of discharge [5]. Headaches disproportionately affect women of childbearing age, and further inequity is aggravated by inaccessibility of newer and high-cost treatments for financially vulnerable patients [4]. Many patients cannot tolerate traditional pharmacotherapy and/or prefer drug-free treatment options. This has led to the development and FDA approval of five new neuromodulation devices to treat migraine; but most are not approved for CM [6]. The FDA requires sham-controlled trials to demonstrate the effectiveness of new headache treatment devices for approval, so well-validated sham devices are critical to bring new noninvasive treatments to the market.

Developing valid sham controls can be difficult for noninvasive devices since subjects can perceive the active signals, which means that subjects perceive the sham signal. The simple act of delivering a signal introduces placebo effect bias. The International Headache Society (IHS) recommends that strategies to maintain blinding should be employed whenever possible when studying neuromodulation therapies and that blinding should be confirmed at the end of each study [7]. Several device-specific sham approaches have been tested, but prospective validation of sham controls is rarely reported. Recently, De Icco et al. mimicked the tingling feeling of anodal direct current stimulation in patients without providing a therapeutic effect in their sham arm by stimulating it only during the first and last 30 s of a 20 min treatment session [8]. A pivotal trial of a noninvasive Vagal Nerve Stimulator (nVNS) (GammaCore Sapphire™, Electrocore, Rockaway, NJ, USA) reported high placebo effect rates, which led to a follow-up study indicating that the sham device significantly modulated the trigeminal autonomic reflex and therefore was not a true sham but rather a partially active device [9].

A novel form of neuromodulation that utilizes cyclical temperature profiles has recently shown promising results. Noninvasive thermal nerve block devices relieved pain from chronic headaches including CM for subjects who started with average baseline pain scores of 6.1 ± 1.7, which decreased to 2.6 ± 2.5 following a single treatment [10]. Zhang et al. showed that a reversible thermal nerve block can be achieved by briefly heating mammalian nerves to 50 °C and then cooling them to room temperature (15–30 °C) [11]. Unmyelinated C fibers can be selectively blocked without blocking Aβ fibers by heating the nerve to ≤45 °C, which allows the block of pain signal transmission without affecting motor function or other minor sensations [12]. Blocking nerve conduction via heating and cooling can be achieved using noninvasive devices for nerves about one-quarter to one-half inch below the skin surface, which includes the occipital and other craniofacial nerves implicated in headache. In addition, noninvasive thermal nerve blocks allodynic scar pain caused by foot surgery as well as pain secondary to trauma of the foot and ankle [13].

The objective of this study was to prepare for a larger sham-controlled pivotal study by confirming that each of the two sham thermal nerve block devices studied could successfully convince subjects that they were undergoing active investigational treatment and that the devices do not provide an active treatment effect. This study builds on previous work by validating both potential sham thermal nerve block devices in a prospective, single-blind trial (Figure 1). Changes in pain levels, credibility of treatments and blinding were assessed during two office visits. The validated sham temperature profiles will be used in future randomized, double-blind sham-controlled trials to demonstrate the efficacy of noninvasive thermal nerve block devices.

## 2. Methods

### 2.1. Inclusion and Exclusion Criteria

An IRB-approved feasibility study was conducted to validate the sham noninvasive thermal nerve block device as a control arm for future studies. Inclusion criteria were male or female aged 21 or older; completion of baseline forms; self-reported headache, migraine with or without aura or other head and/or neck pain; all medication use must be stable for two weeks prior to the study, and patients must be able to give written informed consent. Exclusion criteria were allergy or intolerance to any substance used in the study; any confounding diagnosis or medical history that in the investigator’s opinion would adversely affect the study participation or subject safety; pregnancy; patient’s intolerance to any substance used in the study; and open or breached skin in the region where treatment would be applied.

### 2.2. Enrollment

Study participants were recruited through word of mouth, provider referrals and the clinical practice of the investigators. Informed consent was obtained as approved by the IRB. All study participants consented in a private setting after being given the opportunity to review the informed consent form and ask any questions they had about the study. Seven participants were enrolled and received one (*n* = 2) or two (*n* = 5) sham device treatments utilizing different temperature profiles in an office visit.

### 2.3. Devices

Two sham devices, each with their own temperature profiles, were tested in the same visit to allow for intra-subject comparison of pain relief, perception of treatment credibility and blinding between the profiles while minimizing in-office visits and travel requirements for subjects.

### 2.4. Treatment

Two subjects received only one sham treatment because of scheduling difficulties. The treatment was applied to target the greater occipital nerve, lesser occipital nerve, and third occipital nerve. Two subjects were not able to remain in the clinic for an entire day and did not receive both sham treatments. Following the in-office sham treatment(s), subjects were given the option of using Sham #2 at home for up to one week to see if repeated use of sham devices resulted in participants reporting lower credibility in the beginning of the active treatment. There was no follow-up beyond the 1-week period to investigate longer-term effects of the sham devices. In cases where both shams were administered, Sham #1 was always given prior to Sham #2 because it was anticipated that subjects may no longer perceive the inactive sham treatment to be active/investigational after experiencing the moderate temperature fluctuations associated with Sham #2. Prior to receiving any treatment, most participants (71%) responded “I do not know” when asked if they believed the treatment would work for them, and the remainder (29%) responded “Yes”. The study population mirrored that of the United States migraine population with about 80% female participants, with ages ranging from 25 to 69 years old, and similar median income levels. Education levels varied from those who did not finish high school to graduate degrees (including a lawyer and teacher). Participants were recruited from urban, suburban, and rural areas. Most subjects (57%) suffered from migraine or headache following a traumatic accident (29%), with the remainder (14%) suffering from cervicogenic headache. A summary of the additional study population characteristics is presented in Table 1.

Unblinded site staff were trained to describe the therapy to participants without using the words “hot” or “cold” to describe any specific treatment and to use neutral terms such as “temperature cycles” to reduce the risk of unblinding. Blinded participants were trained to use sham therapies in the first study visit, which lasted approximately 4.5 to 8 h, depending on if participants evaluated one or both sham devices during their first in-office (outpatient) visit. Sham devices labeled Sham #1 pumped fluid at room temperature with the device’s heating and cooling components disconnected. Sham devices labeled Sham #2 cycled between moderately cool (27 °C) and moderately warm (39 °C) temperatures. Fluid flowed through pads that were snugly held against the posterior scalp and neck regions for each treatment. Five of the subjects received treatment from both sham devices (Sham #1 before Sham #2) in the first visit, and two subjects received treatment with Sham #2 but not Sham #1. Sham #1 was tested on subjects before Sham #2 in all subjects receiving both sham devices because it was believed by the authors that the warm and cool temperatures of Sham #2 would influence subjects’ perception of the room-temperature flowing fluid in Sham #1 and could expose Sham #1 as the sham therapy. Pain scores were measured before, during, and after each in-office treatment. Credibility and patient expectations were assessed at the beginning and end while blinding was assessed at the end of each treatment. Participants were allowed to pause treatment to use the restroom and were provided lunch after their first sham treatment. The use of acute headache medication was permitted and was used by one participant during their treatment with Sham #1. The participant who used acute medication had a high level of education (graduate degree) and said they “strongly believe” that the treatment was active despite receiving no relief. The same participant took a second dose of acute medication directly before their scheduled treatment for Sham #2, and their pain completely resolved. Therefore, this subject’s pain scores for Sham #2 were not included in any analysis.

Pain scores were measured using the Defense and Veterans Pain Rating Scale (DVPRS), a pain assessment tool enhanced by functional real-world descriptions, color coding, and pictographs to measure both sham and active treatments as seen in Table 2. The modified expectancy/credibility questionnaire [14], which has previously been used to assess sham headache treatments [15], was administered at the beginning and end of each in-office treatment. Participants were asked the following questions rated on a 0 to 9 scale:

How logical does this type of therapy seem to you for helping people treat or prevent their headache?How confident are you that this treatment will be successful in reducing your headache symptoms if you were to have approximately seven such treatments over the course of one week?How confident would you be in recommending this treatment to a friend who has a problem with headaches?How competent do you consider the individual who gave you the treatment to be?

Question 1 assesses how credible the participant thinks the treatment is; Question 2 assesses subject expectations; and Question 3 quantifies how credible the participant feels the treatment is [14].

Blinding was assessed as recommended by the International Headache Society’s guidelines for clinical trials with neuromodulation devices for the treatment of migraine [7]. To confirm blinding success, a 5-point Likert scale was given at the end of each treatment. James Blinding Index was calculated to assess blinding success and Bang’s Blinding Index (BI) was used as a sensitivity analysis. James Blinding Index ranges from 0 to 1, where 1 signifies perfect blinding (no participants know), 0.5 signifies random guessing, and 0 represents being completely unblinded. Bang’s Blinding Index (BI) ranges from −1 to 1 [16,17], with 1 being a complete lack of blinding (all participants guessed correctly), 0 representing perfect blinding (random guessing), and −1 meaning that all responses were incorrect. Ideal blinding scenarios are either random guesses or participants believing both treatment types are active [18]. At the end of each in-office treatment, participants were asked to choose one of the following statements:I strongly believe treatment is a sham;I somewhat believe treatment is a sham;I do not know if the treatment is a sham or investigational;I somewhat believe the treatment is investigational;I strongly believe the treatment is investigational.

## 3. Results

### 3.1. Response to Blinding

All study participants (100%) believed that both the sham and active treatments were active (investigational) therapies, including 80% of patients receiving sham treatments who “strongly believe” and 20% who “somewhat believe” the treatment was active (Figure 2). The James Blinding Index was 0.5 and Bang’s Blinding Index was 0, both of which represent perfect blinding.

### 3.2. Responses to Credibility and Expectancy Questionnaires

Results from the credibility/expectancy questionnaire are displayed in Figure 3 and show that the average response to each question increased from the beginning to the end of each treatment. All responses to each question had overlapping confidence intervals (within 1 standard deviation), and the difference between responses was less than 2 standard deviations for all but the first question. Question #1 had the largest difference between sham and active treatments, with responses ranging from 5.4 ± 3.6 for Sham #1 at the beginning of treatment up to a maximum of 8.8 ± 0.5 for the active treatment. All subjects whose pain scores were included in the primary analysis provided the same response for all questions for both Sham #1 and Sham #2.

### 3.3. Pain Score Changes before and after Treatment

The reduction in DVPRS pain scores was highest with active treatment (Figure 4), with average reductions of 1.2 for Sham #1, 1.9 for Sham #2, and 3.4 for the active treatment. When data from subjects who took triptans or fell asleep during Sham #2 treatment were excluded, the reduction in DVPRS pain scores was reduced to 1. Two subjects who fell asleep during treatment with Sham #2 were excluded from the primary pain analysis because sleep is considered a migraine treatment, and one additional subject’s data were excluded from Sham #2 pain analyses because they took acute medication minutes before treatment. When the pain scores from subjects who fell asleep were excluded, the pain scores were reduced from 6.4 ± 2.1 to 5.2 ± 2.8 with Sham #1 and from 6.5 ± 2.6 to 5.2 ± 3.2 with Sham #2. Pain was reduced from 5.7 ± 2.5 to 3.5 ± 3.7 with Sham #2 when all subjects were included in the analysis. Pain was reduced from 6.7 ± 2.1 to 2.2 ± 2.5 with active treatment. When all sham treatments were pooled, the baseline pain scores of 6.0 ± 2.2 were reduced to 4.3 ± 3.3. When data from the participants who fell asleep were removed, the average pain reduction from sham treatments was 1.2 ± 1.7. Both shams had reductions in pain of under two points on the DVPRS pain scale with a reduction of 1.2 ± 1.8 seen with Sham #1 and 1.3 ± 1.9 with Sham #2 (1.9 ± 2.1 including all subjects). Active treatment saw a reduction of 3.4 ± 2.5. Additionally, no adverse events were reported in the study.

## 4. Discussion

This study highlights the validation of sham devices prior to their use in large, controlled trials. Testing and validation are critical to confirm successful binding and to minimize the placebo effect of sham devices. Confirming that the sham devices successfully blind study participants without producing active effects reduces the need to administer sham treatment to participants overall by ensuring that future studies are conducted correctly and methodologically sound. Our group previously demonstrated significant pain reduction using active noninvasive thermal nerve blocks in a highly refractory population. In this feasibility study, subjects reported an average pain reduction of 58%, and half of the participants completed the treatment with an NRS of 1 or less [10]. Single-armed feasibility studies are helpful to understand the presence of a treatment effect compared to baseline, but one limitation is the lack of a comparison sham treatment to control for the placebo effect. Sham design is challenging in neuromodulation with a therapy that is associated with perception as compared to sub-perception treatment, as is increasingly used in spinal cord stimulation. Interestingly, Al-Kaisy et al. reported the results of a double-blind randomized crossover study comparing the effects of sham, 1200 Hz, 3030 Hz, and 5882 Hz spinal cord stimulation, in which sham stimulation produced similar analgesic effects of 1200 Hz and 3030 Hz [19].

The sham response seen in this study was within expectations based on the literature. But as demonstrated in Figure 4, the effect of outliers and the need to maintain a consistent subject environment during controlled experiments are crucial. Two participants fell asleep during their in-office sham treatments, and one remarked that they usually experience headache exacerbations caused by loud noises at home. Participating in the study provided them with quiet time away from headache triggers at home, which allowed them to fall asleep and wake up with no pain. Both subjects’ headaches recurred upon returning home, highlighting the importance of maintaining consistent subject environments and levels of alertness. With the results from subjects who fell asleep omitted, starting pain levels were equivalent for all sham and active treatments, indicating that subjects’ pain had returned to baseline before Sham #2 (after receiving Sham #1 earlier on the same day) and also before active treatment (administered one week later). In fact, baseline pain levels were nearly identical on average between all groups. When outliers with confounding influences were excluded, the average sham pain reduction decreased by half a point on the DVPRS scale, and both shams showed similar pain reduction. These results highlight the importance of maintaining consistent environments for all subjects and preventing subjects from falling asleep during headache trials.

Responses to the blinding questions show an ideal blinding scenario as described by Park et al. [20], where all sham and active group participants in this study believed that their treatments were active. It is likely that patients receiving sham treatments hoped that they were receiving active treatment. Several patients stated verbally that they “knew” the sham devices were active treatment because they could feel the device pumping fluid even when room-temperature fluid was used. One patient wrote a letter to thank us for the chance to participate and said that they would recommend the sham treatment to friends and family even though it did not work for them.

The objective of this study was to help finalize the design of a future pivotal study, and the preliminary results from this study confirm plans for a sham-controlled pivotal trial. Due to the small sample size and the inability to prospectively conduct the study without the preliminary data generated from it, the efficacy results should not be generalized. However, the pain reduction from sham treatments was consistent with other reports of noninvasive sham headache treatments [21], which gives us confidence to prospectively conduct a future study to demonstrate efficacy. For a future study that allocates patients 1:1 into sham and active arms with 90% study power and α = 0.05, 36 subjects would be needed using Sham #1 (and also for Sham #2, excluding the subjects who fell asleep), though 76 subjects would be needed using Sham #2 if all subjects’ pain relief data were included. More than twice as many subjects would be needed if environmental conditions and subjects’ level of alertness cannot be adequately controlled, highlighting the critical importance of consistency during trials. Pain reduction from both sham therapies was consistent with results seen using other acute sham external trigeminal nerve stimulation treatments [21]. Chou et al. reported a VAS reduction of 2.6 ± 1.9 from sham Cefaly treatments [21], which overlaps with the DVPRS reductions seen with both thermal nerve block devices, Sham #1 and Sham #2. The small sample size of the study and short study duration limit interpretation of the long-term efficacy of the treatment.

In a sham device validation study, the confirmation of subject blinding is fundamental. Our assessment of blinding was conducted using the blinding index proposed by Bang et al. [17] In this paper, a blinding index was scaled in an interval of −1 to 1, with 1 being complete lack of blinding, 0 being perfect blinding, and 1 indicating opposite guessing (unblinding). Our study had a blinding index where 0 equates to perfect blinding per the method proposed by Bang et al. Another tool used to assess blinding is the James Blinding Index [22]. In this index, the scale has an interval of 0 to 1, with 0 being a complete lack of blinding and 1 being complete blinding [22]. Our study had a James Blinding Index of 0.5, which again validates the extent of subject blinding in this study.

Current guidelines by the International Headache Society (IHS) written by Tassorelli et al. (as well as a member of our group) discuss clinical trials with devices for migraine treatment [7]. A placebo (sham) is defined as “Control group uses or is exposed to another or same device that is externally indistinguishable from the active device but whose stimulation is believed to have no therapeutic effect” [7]. The purpose of a placebo (sham) control is to control for the placebo effect. Blinding is intended to ensure that subjective assessments and decisions are not affected by knowledge of treatment assignment. According to the FDA, effective blinding is essential to maintain the reliability of trial findings and the study’s capacity to account for the placebo effect. Participants in our study believed that they were truly receiving treatment with the sham devices. Based on verbal comments made by participants, we hypothesize that the subjects felt the fluidic flow and vibration of the sham device and interpreted these to be part of the active therapeutic effect of a real investigational device. The subjects then perceived that they were undergoing active therapy. Additionally, the credibility of the sham device lasted beyond their initial use. When research coordinators followed up with participants to schedule their next study visit, study participants continued to report credibility over the week that the sham treatment was investigational despite the lack of any actual efficacy. Our results with this phase of the study are extremely important as future randomized controlled clinical trials will consist of long-term treatment for 12 weeks as recommended by the IHS, and credibility can play a significant role in long-term clinical improvement [15,23,24,25,26].

An important point to note is the ethical use of sham devices in a research setting. As discussed by Miller et al., there is a sense of moral discomfort that may occur during investigative trials utilizing sham devices. In these situations, an important point to consider is that in many cases, sham procedures follow the same principles used with common research interventions and does not violate the rights of patient subjects as long as an informed consent process has taken place. The informed process allows the patient to understand the presence of both an active treatment and a sham and have the ability to make their own decision [27].

## 5. Conclusions

This study demonstrated that two potential sham thermal nerve block devices could be used for future randomized controlled trials. Sham devices, with heating and cooling electronics disconnected, showed equivalent placebo responses compared to devices that moderately heated and cooled, and both devices saw pain reduction similar to that observed in literature reports for other noninvasive sham neuromodulation devices. All study participants thought that each sham device was an active investigational device, and credibility was the same (within one standard deviation) for all sham and active devices, indicating that either device may be suitable for use as a control in future controlled trials.

## Figures and Tables

**Figure 1 brainsci-13-01718-f001:**
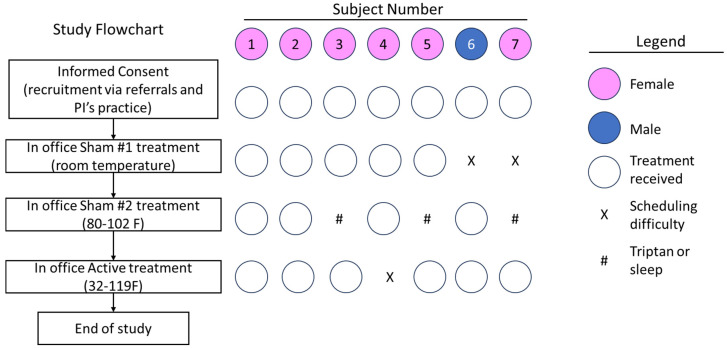
Trial flow diagram. Pink circles represent female participants, blue circles represent the male participant, X indicates the treatment was not conducted because of scheduling difficulties, and # indicates that the patient’s pain scores were excluded from the primary analysis due to taking a triptan (subject #3) or falling asleep (subjects 5 and 7).

**Figure 2 brainsci-13-01718-f002:**
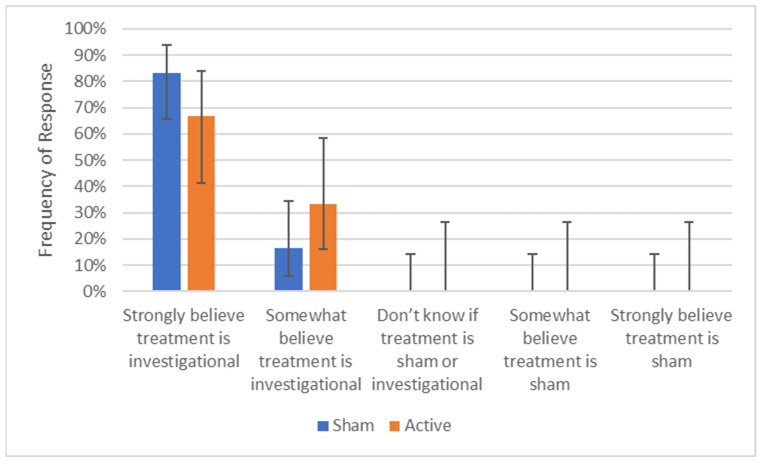
Responses to 5-point blinding scale. Error bars show one standard deviation.

**Figure 3 brainsci-13-01718-f003:**
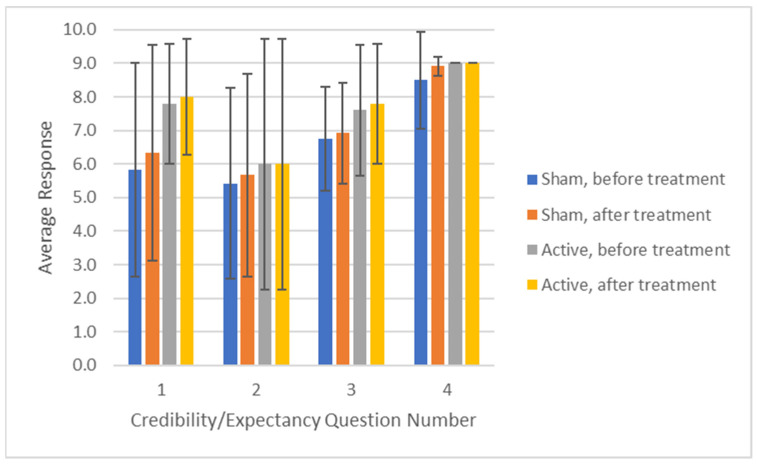
Responses to the credibility/expectancy questionnaire administered at the beginning and end of each in-office treatment. Error bars show one standard deviation.

**Figure 4 brainsci-13-01718-f004:**
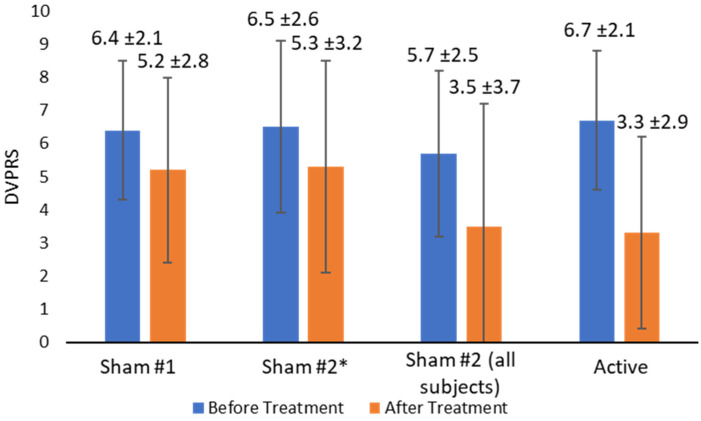
Average DVPRS pain scores before sham and active treatment. Error bars show one standard deviation. * excludes subjects who fell asleep (*n* = 2) or used acute headache treatment (*n* = 1).

**Table 1 brainsci-13-01718-t001:** Sham validation study patient characteristics.

Characteristic	Study Population
% Female	85%
BMI	30.4 ± 7.0
Age	45.3 ± 16.9
% Chronic migraine	57%
% Post-traumatic headache	29%
% Cervicogenic headache	14%
% Current triptan users	43%
Baseline pain score	6.5 ± 2.0
Headache days	
Past 7 days	5.7 ± 2.2
Past 28 days	21.9 ± 8.8

**Table 2 brainsci-13-01718-t002:** Average and standard deviation of responses to the credibility/expectancy questionnaire administered at the beginning and end of each in-office treatment.

Credibility Question	1	2	3	4
Before Sham #1	5.4 ± 3.6	5.2 ± 2.9	6.6 ± 1.1	8.8 ± 0.4
After Sham #1	6.0 ± 3.5	5.0 ± 2.9	6.6 ± 1.1	8.8 ± 0.4
Before Sham #2	6.1 ± 3.1	5.6 ± 3.0	6.9 ± 1.9	8.3 ± 1.9
After Sham #2	6.3 ± 3.3	6.1 ± 3.2	7.4 ± 1.7	9.0 ± 0.0
Before Sham (combined)	5.8 ± 3.2	5.4 ± 2.8	6.8 ± 1.5	8.5 ± 1.4
After Sham (combined)	6.2 ± 3.2	5.7 ± 3.0	7.1 ± 1.5	8.9 ± 0.3
Before Active	8.5 ± 1.0	6.3 ± 4.3	8.3 ± 1.5	9.0 ± 0.0
After Active	8.8 ± 0.5	6.3 ± 4.3	8.5 ± 1.0	9.0 ± 0.0

## Data Availability

The data that support the findings of this study are available on reasonable request from the corresponding author. The data are not publicly available due to privacy/ethical restrictions.

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
