# Peer review of "Drug-Free Noninvasive Thermal Nerve Block: Validation of Sham Devices"

_brainsci, 2023, doi:10.3390/brainsci13121718_

Round 1
Reviewer 1 Report
Comments and Suggestions for Authors
Dear editor,
Thank you very much for giving me opportunity to read and assess the entitled article “ Drug-Free Noninvasive Thermal Nerve Block: Validation of Sham Devices” which was submitted your esteemed journal.
Firstly, I congratulate the authors for their study and efforts. I think their opinion for the study is very valuable. However, I think there are some problems in the methodology.
My suggestions;
I think sample size is very low. İf it is possible, I suggest them to increase study population.
I suggest to authors to preapere the flow-chart and explain the method of the study more clearly for readers.
it is not clear why the authors only practiced sham therapy and did not include a control or investigation group?
What is the hypotesis of this study?
I think this is not clear “Seven participants were enrolled and received one (n=2) or two (n=5) sham device treatments utilizing different temperature profiles in an office visit.”
Why 2 patients received one time? And others received two times?
Congratulate the authors on the basis of the working question, I believe that the authors could have composed the article with a stronger methodology and presented it as a higher quality research.
Comments on the Quality of English LanguageMinör editing
Author Response
Firstly, I congratulate the authors for their study and efforts. I think their opinion for the study is very valuable. However, I think there are some problems in the methodology.
My suggestions;
I think the sample size is very low. İf it is possible, I suggest they increase their study population.
Unfortunately, at this time, we are unable to increase the size of the study population as this paper and study has completed.
I suggest to authors to prepare the flow-chart and explain the method of the study more clearly for readers.
Figure 1 already shows the methodology of the study and the number of participants included in each step of the study
it is not clear why the authors only practiced sham therapy and did not include a control or investigation group?
Our goal of this paper was to validate the sham device and show that patients believed they were receiving active treatment
What is the hypothesis of this study?
Our hypothesis was that the sham device used in the study is validated and the participants receiving the sham treatment will believe that they are getting active treatment as described in the abstract and in the paper.
I think this is not clear “Seven participants were enrolled and received one (n=2) or two (n=5) sham device treatments utilizing different temperature profiles in an office visit.” Why 2 patients receive one time? And others received two times?
two subjects only received one of the shams because of scheduling difficulties. We have included that reasoning into the paper.
Congratulate the authors on the basis of the working question, I believe that the authors could have composed the article with a stronger methodology and presented it as a higher quality research.
Thank you for your comments
Reviewer 2 Report
Comments and Suggestions for Authors
Interesting paper. The responsiveness of peripheral nerves to alternating thermal stimuli resulting in nerve blockade has an underlying mechanism where temperature mediated changes in pain are often observed. To assess the real efficacy of this treatment we need to compare it in multiple randomized controlled trials in migraine population as reported in use of devices guidelines, so the importance of validating electronic safe sham devices.
The study is pivotal and a small number of migraine people tested it so further studies are needed to confirm the validity and the safety. I have only a suggestion for the authors that is to add if any adverse event occurs or not during the study.
Author Response
Thank you for your comments. We have included a comment about the adverse events during the study which were no adverse events.
Reviewer 3 Report
Comments and Suggestions for Authors
· The manuscript lacks information on participant recruitment and the informed consent process. It is essential to include details on how participants were selected, their eligibility criteria, and how informed consent was obtained, ensuring ethical considerations are met.
· The method section is completely vague kindly write it in multiple subheadings for better clarity.
· The study mentions that active treatments showed sustained pain relief during optional at-home use, but the duration of this relief is not elaborated. Were there any follow-up assessments beyond the optional week of at-home use?
· While the study reports the credibility of both sham and active treatments, more information is needed about the criteria and scales used to measure credibility. How were these assessments conducted, and what specific factors were evaluated?
· The paper should discuss ethical considerations in greater depth, particularly the ethical implications of using sham devices. This is crucial to address potential concerns regarding the use of placebos in medical research.
· Ensure that all claims and statements are properly supported with relevant references and citations.
· The manuscript should undergo a thorough language and clarity check. Some sentences are long and complex, making it challenging to follow the logic. Simplify the language for better comprehension.
· The manuscript mentions the use of Bang's Blinding Index and James' Blinding Index to confirm successful blinding, but no further details or results are provided. Include the actual values of these indexes and explain their interpretation
· Authors should write the proper subheadings for the results section as in the current form the message is not clear.
· Include error bars in all the figures
Author Response
- The manuscript lacks information on participant recruitment and the informed consent process. It is essential to include details on how participants were selected, their eligibility criteria, and how informed consent was obtained, ensuring ethical considerations are met.
The methods section has been included to talk about the IRB study that was obtained. Additionally, more details have been included to discuss the setting of the informed consent process and the step by step process of enrollment.
- The method section is completely vague, kindly write it in multiple subheadings for better clarity.
Subheadings have been used to separate the methods section
- The study mentions that active treatments showed sustained pain relief during optional at-home use, but the duration of this relief is not elaborated. Were there any follow-up assessments beyond the optional week of at-home use?
Revised the methods section to clarify that there was no follow-up to investigate sham effects beyond 1 week of at-home use
- While the study reports the credibility of both sham and active treatments, more information is needed about the criteria and scales used to measure credibility. How were these assessments conducted, and what specific factors were evaluated?
The manuscript mentions the use of Bang's Blinding Index and James' Blinding Index to confirm successful blinding, but no further details or results are provided. Include the actual values of these indexes and explain their interpretation
Blinding was assessed as recommended by the International Headache Society’s guidelines for clinical trials with neuromodulation devices for the treatment of migraine.7 To confirm blinding success, a 5-point Likert scale was given at the end of each treatment. James Blinding Index was calculated to assess blinding success and Bang’s Blinding Index (BI) was used as a sensitivity analysis
- The paper should discuss ethical considerations in greater depth, particularly the ethical implications of using sham devices. This is crucial to address potential concerns regarding the use of placebos in medical research.
Ethicality on the use of sham devices has been discussed as well as references
- Ensure that all claims and statements are properly supported with relevant references and citations.
All references have been included and labeled
- The manuscript should undergo a thorough language and clarity check. Some sentences are long and complex, making it challenging to follow the logic. Simplify the language for better comprehension.
Several sentences have been changed to ensure more clarity.
- Authors should write the proper subheadings for the results section as in the current form the message is not clear.
Subheadings have been added to the paper
- Include error bars in all the figures
- Added error bars to figures 2 and 3 and updated their captions
Round 2
Reviewer 1 Report
Comments and Suggestions for Authors
The methodology of the study still does not specify how these patients were included in the study, the population and sample of the study.
Have these patients been admitted somewhere?
Thank you
Comments on the Quality of English LanguageAppriopriate
Author Response
Hello, new reviews and revisions have been addressed and new version of the paper has been uploaded